# Stability across the Whole Nuclear Genome in the Presence and Absence of DNA Mismatch Repair

**DOI:** 10.3390/cells10051224

**Published:** 2021-05-17

**Authors:** Scott Alexander Lujan, Thomas A. Kunkel

**Affiliations:** Genome Integrity Structural Biology Laboratory, National Institute of Environmental Health Sciences, National Institutes of Health, DHHS, Research Triangle Park, NC 27709, USA; lujans@niehs.nih.gov

**Keywords:** mutation accumulation, mutagenesis, mutation rate, DNA replication, genome stability, DNA mismatch repair, whole-genome sequencing, DNA repair, Eukarya

## Abstract

We describe the contribution of DNA mismatch repair (MMR) to the stability of the eukaryotic nuclear genome as determined by whole-genome sequencing. To date, wild-type nuclear genome mutation rates are known for over 40 eukaryotic species, while measurements in mismatch repair-defective organisms are fewer in number and are concentrated on *Saccharomyces cerevisiae* and human tumors. Well-studied organisms include *Drosophila melanogaster* and *Mus musculus*, while less genetically tractable species include great apes and long-lived trees. A variety of techniques have been developed to gather mutation rates, either per generation or per cell division. Generational rates are described through whole-organism mutation accumulation experiments and through offspring–parent sequencing, or they have been identified by descent. Rates per somatic cell division have been estimated from cell line mutation accumulation experiments, from systemic variant allele frequencies, and from widely spaced samples with known cell divisions per unit of tissue growth. The latter methods are also used to estimate generational mutation rates for large organisms that lack dedicated germlines, such as trees and hyphal fungi. Mechanistic studies involving genetic manipulation of MMR genes prior to mutation rate determination are thus far confined to yeast, *Arabidopsis thaliana*, *Caenorhabditis elegans*, and one chicken cell line. A great deal of work in wild-type organisms has begun to establish a sound baseline, but far more work is needed to uncover the variety of MMR across eukaryotes. Nonetheless, the few MMR studies reported to date indicate that MMR contributes 100-fold or more to genome stability, and they have uncovered insights that would have been impossible to obtain using reporter gene assays.

## 1. Overview

We considered 123 independent nuclear genome mutation rate measurements, gathered from over 90 studies performed in either wild-type or MMR-deficient strains of 48 eukaryotic species. We confined the analysis to whole-genome studies that explicitly report rates or, rarely, to studies from which rates may be easily calculated. We present mean mutation rates for similar systems that are either MMR-proficient (Table 1) or MMR-deficient (Table 2). Studies are listed in Table 3. Granular details and notes on each study may be found in Appendix A. Where available, rates per generation and per cell division are both presented. We classify each estimate as using either germline or somatic cells, although there is no distinction for most unicellular eukaryotes, and some organisms, e.g., hyphal fungi and many plants, lack dedicated germlines. We highlight trends and extremes, and then comment on how whole-genome rates elucidate mechanisms of MMR.

## 2. A Brief History

Mutation accumulation (MA) experiments are a venerable approach for estimating spontaneous mutation rates (reviewed in [1]). Theorized in the 1920s and first implemented in the 1960s, MA experiments use replicate lines derived from an ancestral population that can evolve independently. The population is subjected to periodic artificial bottlenecks to fix mutations regardless of their effects on selective fitness. Originally, mutations were selected via phenotypic changes due to mutations in reporter loci. Sequencing a reporter locus in the final population allowed for mutation detection and counting, resulting in mutation spectra and mutation rate estimates for that locus. However, no reporter locus can simulate all possible contexts, transcription states, chromatin states, replication times, or proximity to various genomic features. The advent of whole-genome sequencing bypassed these restrictions by making the entire genome the reporter. Mutations may be called by comparing the parental sequence to the sequences of progeny populations.

The first successful whole-genome MA experiments were published in 2008, first in *Saccharomyces cerevisiae* (baker’s yeast) [2] and then in the bacterium *Salmonella typhimurium* [3]. Wild-type, whole-genome mutation rates were previously compared across kingdoms [1], and therefore here we confine the discussion to the eukaryotic nuclear genome. Lynch et al. found 33 mutations in four wild-type haploid *Saccharomyces cerevisiae* lines that had been each propagated for approximately 4800 cell divisions [2]. They estimated the whole-genome mutation rate at 0.33 Gbp^−1^ division^−1^. The race was on to find rates in as many diverse species as possible. By the end of 2010, the list included model organisms such as *Drosophila melanogaster* (fruit or vinegar fly; 0.1 Gbp^−1^ division^−1^; [4]), *Caenorhabditis elegans* (a roundworm; 0.32 Gbp^−1^ division^−1^; [5]), and *Arabidopsis thaliana* (thale cress; 0.22 Gbp^−1^ division^−1^; [6]).

The first whole-genome mutation rate estimates for genetically manipulated eukaryotes were also published in 2010. Zanders et al. performed the first estimates for DNA mismatch repair (MMR)-deficient organisms, a baker’s yeast strain with a temperature-sensitive variant of the MMR gene *MLH1* (*mlh1-7^ts^*; 3.7 Gbp^−1^ division^−1^ [7]). Comparison with the wild-type rates of Lynch et al. implied MMR repair of over 90% of replication errors (MMR–/MMR+ = correction efficiency; 3.7/0.33 = 11.2). This comported well with prior reporter locus estimates in [8]. Larrea et al. then used MMR-deficient (*msh2*Δ) baker’s yeast with a variant of DNA polymerase (Pol) δ (*pol3-L612M*; [9]). The known mutation bias of *pol3-L612M*, found in previous experiments in vitro [10], showed the bulk of Pol δ synthesis to occur on the nascent lagging strand. This extended results from previous mutation accumulations in reporter genes [11]. Thus, whole-genome mutation collections were shown to be useful for revealing cellular mechanisms.

A study in 2010 also reported the first whole-genome mutation rate estimate for humans (11 Gbp^−1^ generation^−1^ [12]). This estimate could not come from whole-genome MA experiments. Baker’s yeast can reproduce through budding, a form of binary fission. Baker’s yeast, roundworms, and thale cress can reproduce through selfing. Vinegar flies neither bud nor self-fertilize, but they can be inbred in order to fix mutations. None of these options are available for humans, and therefore Roach et al. sequenced the genomes of a nuclear family and inferred mutations by comparing children to parents. Such parent-offspring sets are now a standard method for finding whole-genome mutation rates in outcrossing species, including wild populations.

The following decade saw scores of whole-genome rate measurements, plus a host of mutation frequencies and spectra from tumor genomes (e.g., [13]). Note that tumor studies often use similar terminology and technology to the experiments listed here, but, lacking cell division counts, they may report mutation frequencies rather than rates. This restriction has been circumvented somewhat by measurements in cancer cell lines (e.g., chicken DT40 tumor line [14,15] and human cell line RPE1 [16], and by raising organoids from tumor samples [17]). The latter is also useful for estimating mutation rates in normal somatic tissues [17,18].

Some progress has been made in calculating mutation rates given incomplete knowledge of ancestral states or generation counts. Where complete pedigrees are unknown or ancestral samples are unavailable but little selective pressure is expected, mutations may be inferred by deriving the genotype of the last common ancestor. This technique, known as identity by descent, limits analysis to certain highly conserved segments [19]. Likewise, the number of cell divisions in the stem line for a particular tissue may be unknown. Given a representative sample of the whole tissue, the mutation rate in the first few rounds of replication may be inferred from variant allele frequencies (VAF). VAF methods are easiest with blood and require sophisticated modelling to account for unequal contributions of early embryonic cells [20].

## 3. Nuclear Mutation Rates in MMR-Proficient Germ Cells

Mutation rates are, by necessity, conditional. There is little ab initio reason to expect mutation rates to remain constant across differing species, environmental conditions, stressors, exposures, tissues, and germline versus somatic status. For instance, mutation rates may vary with organismal, tissue, or parental ages. Human mutation counts increase with parental age, particularly paternal age, which affects the mutation rate per generation [21]. Wherever necessary, Appendix A uses assumed average parental age, as defined by the authors of the study in question. Somatic rates are averaged across estimates, including across tissues [18] and growth conditions [22], where rates vary little. Rates are not combined if conditions are known to cause large rate differences, such as different ploidies [23] or homozygous versus hybrid or otherwise highly heterozygous individuals [24]. Mutation rates are also conditional across individual genomes, a subject we address below in our discussion of MMR.

Wild-type generational mutation rates range from 0.00761 Gbp^−1^ generation^−1^ in the ciliate *Tetrahymena thermophila* [25] to 3380 Gbp^−1^ generation^−1^ in the hyphal fungus *Neurospora crassa* (red bread mold; [26]). These extremes are largely explainable by how these organisms transmit their genetic code through generations. Ciliates such as *Tetrahymena* and *Paramecium tetraurelia* [27] keep dozens of working copies of their genome in transcriptionally active compartment called the macronucleus while protecting a germ copy in a protected micronucleus. In contrast, red bread mold has no separate germ line, undergoing an average of 300 asexual divisions per sexual generation [26]. The asexual rate is listed as “somatic” in Table 1, although this definition is debatable. However, for reasons that are not entirely clear, most mutations per sexual generation occur in the last few divisions, perhaps only during meiosis. Is this the case with meiosis in other organisms?

Another hyphal fungus, the fairy ring mushroom *Marasmius oreades*, has the lowest measured mutation rate per cell division at 0.0038 Gbp^−1^ division^−1^ [26]. This is even lower than in the ciliates, but without an obvious mechanistic explanation. Nonetheless, with over 19,000 divisions per generation, this still yields a relatively high mutation rate of 73 Gbp^−1^ generation^−1^. How is the low rate per division maintained and how is the high rate per generation tolerated? Is this situation common among hyphal fungi? The cell divisions per generation for the fairy ring mushroom in Table 1 are estimated from the ratio of rates, per-generation divided by per-division. This would be incorrect if it has a sexual rate dominated by mutations in later cell divisions, as in red bread mold. Perhaps clarity will emerge through testing more organisms with more diverse lifestyles and genetic architectures. The highest wild-type “germline” mutation rate per division is 0.98 Gbp^−1^ division^−1^, in the haploid unicellular alga *Micromonas pusilla* [28]. How does this organism deal with a rate per division over 250-times higher than in the fairy ring mushroom? This rate is in turn dwarfed by those in animal somatic cells.

## 4. Nuclear Mutation Rates Trends in MMR-Proficient Organisms

Three trends in nuclear mutation rates appear in the data. First, as previously stated for humans, mutation rates increase with parental age. Second, in plants, highly heterozygous lines have higher mutation rates than homozygous lines. Third, in animals, somatic mutation rates exceed germline rates. Mutation counts increase with parental age in many species. In humans, paternal age has a particularly strong effect on offspring mutation counts, commensurate with continuing cell divisions in the male germline (reviewed [21]). However, maternal age is also a factor, which is more difficult to explain. Although outside the scope of this review, whole mitochondrial genome sequencing studies also show age-dependent increases in both point mutations [29] and large deletions [30]. The situation is even more extreme in large, long-lived hyphal fungi [31,32] and trees [33,34,35,36,37]. Because they grow outward linearly, lack a dedicated germline, and tend to fruit near their maximum extent, each consecutive fruiting results in more offspring mutations. Will whole-genome mutation rate studies ever find age-related increases in shorter lived or unicellular eukaryotes?

Only two whole-genome mutation rate studies were found that compared homozygous lines with highly heterozygous lines. Both were in plants, encompassing three species. Yang et al. found 3.6-fold higher rated in heterozygous thale cress and 3.4-fold higher rates in heterozygous rice (Oryza sativa) [24]. Likewise, Xie et al. found a more modest 1.6-fold increase in a hybrid peach tree (*Prunus davidiana* × *P. persica*) versus in a weakly heterozygous peach tree (*P. persica*) [33]. Both studies concluded that highly heterozygous lines have higher mutation rates than homozygous lines. The idea that heterozygosity is tied to plant mutation rates has been discussed [33] and is supported by previous reporter locus assays (e.g., [38]). Will the results of these few experiments be recapitulated in other plants or in other eukaryotic clades?

One study measured comparable somatic and germline mutation rates per cell division in two organisms: humans and house mice [39]. The highest measured wild-type mutation rate per cell division belongs to house mouse fibroblasts at 8.1 Gbp^−1^ division^−1^, roughly 70-fold higher than in the germline. Likewise, human fibroblasts rates were 2.7 Gbp^−1^ division^−1^, roughly 80-fold higher than in the germline. Is this a general feature of multicellular organisms other than hyphal fungi, or is it limited to just animals or to mammals only? How are lower mutation rates maintained in the germline? Does MMR play a part or is it only a matter of protection from insult exposure? More information is needed in other animals and multicellular fungi, plants, and stramenopiles (e.g., kelp).

## 5. Nuclear Mutation Rates in MMR-Deficient Cells

Table 2 lists overall mutation rates in MMR-deficient cells. These come from baker’s yeast, fission yeast (*Schizosaccharomyces pombe*), thale cress, roundworms (*C. elegans*), and an immortalized chicken cell line (*Gallus gallus domesticus* DT40). The mean rates have non-overlapping ranges: MMR-proficient with 0.23–0.91 Gbp^−1^ division^−1^, and MMR-deficient with 13–72 Gbp^−1^ division^−1^. Correction efficiencies are remarkably consistent, ranging from 50- to 130-fold, despite disparate species, ploidies, cellular lineages (i.e., somatic versus germline), and methods for ablating MMR (see Appendix A for genotypes and notes). The correction efficiencies are bimodally distributed, with fission yeast, chicken cells, and diploid baker’s yeast clustered at 51–57× and thale cress, roundworms, and haploid baker’s yeast efficiencies from 100–130×. Is this a coincidental artefact of the few systems studied? Regardless, these whole-genome rate measurements have clearly shown that MMR is highly efficient, repairing at least 98% of replication errors. Indeed, this is probably an underestimate (see Section 8).

## 6. Genome-Wide Mutations and the Mechanisms of MMR

For long-lived organisms, reporter locus experiments are an inefficient way to collect mutations. For shorter-lived organisms, given the expense of whole-genome sequencing and the time required for mutation accumulation experiments (ideally hundreds of generations), why not use reporter loci? First, reporter loci do not adequately model the sequence complexity of the genome (as discussed above). Second, reporter loci cannot replicate the diversity of selective pressures across the genome. Both factors are essential for the study of MMR.

For example, the baker’s yeast genome is GC-poor, but certain AT-rich features are concentrated outside of regions that are translated into proteins (like most reporter loci). AT homopolymer tracts, particularly long tracts, are concentrated in untranslated regions (UTRs) that flank most genes [40]. This leads reporter locus assays to underestimate the rates of deletions in long homopolymers and the rates of multi-base insertions and deletions (indels) [41]. Whole-genome mutation accumulations show that these regions become indel hotspots upon removal of MMR [40], with rates and indel sizes increasing with tract length [40,42]. In fact, the shape of the curve of rate versus tract length is diagnostic of the degree to which mismatch extension is favored over proofreading. Extension could be driven by a proofreading defect [43] or by alteration of nucleotide concentrations [44].

Unlike in yeast, AT homopolymers in humans are concentrated in genes, where cancer genomes indicate strong transcriptional strand asymmetry for indels [45,46]. Studies of tumors with Pol δ proofreading defects suggest that MMR repairs about threefold more mismatches produced during lagging strand replication compared with leading [45]. Massive studies of cancer genomes have allowed the construction of mutation spectrum signatures that are diagnostic of such processes as MMR [47,48]. Tumors with mutations in DNA polymerase (Pol) ε have mutation spectra that resemble spectra from cell lines with defects in both Pol ε and MMR [49]. This suggests that MMR is somehow suppressed in those tumors. Conversely, there appears to be a mutational hotspot in the gene that encodes the catalytic subunit of Pol ε in MMR-deficient mouse lymphomas [50]. Spectra in MMR-deficient chicken cells allowed Németh et al. to collapse six MMR-associated COSMIC signatures into two [15]. They found no correlation between these signatures and the identity of the defective MMR genes in the tumors (i.e., *MSH2*, *MSH6*, or *MLH1*). This suggests that either modulation of transcription or translation or some form of inhibition are to blame for the MMR defects in these tumors. This is a profound revelation, given that MMR-deficient cancers generate mutant neoantigens that make them sensitive to immune checkpoint blockade [51]. Thus, whole-genome mutation rate experiments may affect cancer diagnosis and treatment.

Whole-genome experiments have revealed that MMR preferentially protects many genome features. In baker’s yeast, it protects UTRs and inter-nucleosome linkers from indels, translated gene bodies from point mutations, and sequence-encoded nucleosome positions from substitutions [40]. Much of this is recapitulated in thale cress [52], and in humans, MMR selectively protects exons relative to introns [53]. In fission yeast, MMR selectively protects euchromatin [54]. Baker’s yeast strains have slightly higher rates in early as opposed to late replicating regions, with some indication of higher MMR efficiency early in replication [40]. Likewise, variable human MMR is thought to cause elevated mutation rates in late replicating heterochromatin compared to early replicating euchromatin [55]. Are MMR proteins depleted or in some other way impaired later in replication? In humans, some MMR proteins are differentially expressed across the cell cycle [56]. In mice, histone modifications can target MMR to transcriptionally active regions [57], both locally and globally [58]. The extent of targeting elsewhere and in other organisms is unknown. Unfortunately, those these trends point in the same direction, only a few of these studies report rates [15,40,52,54], making it difficult to compare effects across organisms in a quantitative manner.

Why does MMR appear to selectively protect some features over others? Perhaps the extent of MMR targeting, as in mice, is underappreciated. Alternatively, MMR may operate at a similar rate across each genome, but some contexts are simply more mutable. This would be expected if natural selection effectively erases mutations missed by MMR. Over evolutionary timescales, mutable sequences would disappear in regions under little selection. Depletion of MMR would then reveal the fingerprints of past selection (discussed in [40]).

## 7. Summary

Herein, we have gathered known whole-genome mutation rates, encompassing 90 studies (Table 3). We hope that future researchers will expand the list and use the information to uncover new insights into the patterns of mutagenesis across eukaryotes and beyond. We have also outlined some advances in the understanding of mutagenesis since the advent of whole-genome experiments. These advances reveal variation in eukaryotic DNA mismatch repair mechanisms that were invisible to most reporter locus assays. Further progress requires more breadth in the organisms, tissues, and conditions. In particular, new strains are required to uncover the interplay between mismatch repair and other nuclear systems, such as nucleotide pool maintenance, exonucleolytic proofreading, and ribonucleotide excision repair.

## 8. More Future Questions

In addition to questions throughout this review, others arise due to the following. MMR efficiency calculations presented here assume that all mutations are due to replication and are subject to mismatch repair. The veracity of these assumptions is an outstanding question. For instance, most spontaneous mutations in wild-type yeast could be due to mutagenic repair of spontaneous lesions [118], which may not be amenable to MMR. Indeed, 40–85% of mutations in the wild-type baker’s yeast *CAN1* reporter are attributable to errors made by DNA polymerase ζ [119,120,121,122]. Is this true across the genome, in other organisms, other conditions, or in various tissues? How much of the remaining wild-type mutation rate is due to other assumption-breaking processes? Is MMR dependent on other systems, such that a mutation that effects MMR also alters, say, polymerase proofreading or ribonucleotide excision repair, thus causing additional complicating mutagenesis? Until such questions are answered, all MMR efficiency calculations are likely to be minimum estimates and should be treated as provisional.

## Figures and Tables

**Table 1 cells-10-01224-t001:** Nuclear genome mutation rates from whole-genome experiments (MMR-proficient).

						Germ	Mutation Rates			
ct.	Species	Supergroup	Lower Clade	Cellu-Larity	Ploidy	V. Soma	Gbp^−1^ gen.^−1^	Gbp^−1^ div.^−1^	Lines	Mutations
1	*Phaeodactylum tricornutum*	TSAR Group	Stramenopiles	uni-	2n	g	0.49	0.49	36	156
1	*Paramecium tetraurelia*	TSAR Group	Ciliophora	uni-	2n	g	0.030	0.03	7	29
1	*Tetrahymena thermophila*	TSAR Group	Ciliophora	uni-	2n	g	0.0076	0.0076	8	5
1	*Plasmodium falciparum*	TSAR Group	Apicomplexa	uni-	1n	g	0.25	0.25	279	85
1	*Bathycoccus prasinos*	Archaeplastida	Chlorophyta	uni-	1n	g	0.44	0.44	37	32
3	*Chlamydomonas reinhardtii*	Archaeplastida	Chlorophyta	uni-	1n	g	0.18	0.18	91	6890
1	*Micromonas pusilla*	Archaeplastida	Chlorophyta	uni-	1n	g	0.98	0.98	36	85
1	*Ostreococcus mediterraneus*	Archaeplastida	Chlorophyta	uni-	1n	g	0.59	0.59	37	65
1	*Ostreococcus tauri*	Archaeplastida	Chlorophyta	uni-	1n	g	0.48	0.48	40	104
5	*Arabidopsis thaliana*	Archaeplastida	Embryophyta	multi-	2n	g	6.7	0.26	156	2324
1	*Arabidopsis thaliana*	Archaeplastida	Embryophyta	multi-	2n (het.)	g	27	-	99	299
1	*Eucalyptus melliodora*	Archaeplastida	Embryophyta	multi-	2n	g	62	-	1	90
1	*Lemna minor*	Archaeplastida	Embryophyta	multi-	2n	g	0.087	-	16	29
1	*Oryza sativa*	Archaeplastida	Embryophyta	multi-	2n	g	3.2	-	5	10
1	*Oryza sativa*	Archaeplastida	Embryophyta	multi-	2n (het.)	g	11	-	11	55
1	*Picea sitchensis*	Archaeplastida	Embryophyta	multi-	2n	s	27	-	20	5
1	*Populus trichocarpa*	Archaeplastida	Embryophyta	multi-	2n	g	2.0	-	2	186
1	*Prunus hybrid*	Archaeplastida	Embryophyta	multi-	2n (het.)	g	14	-	30	171
1	*Prunus persica*	Archaeplastida	Embryophyta	multi-	2n	g	8.6	-	32	114
1	*Quercus robur*	Archaeplastida	Embryophyta	multi-	2n	s	47	-	1	17
1	*Silene latifolia*	Archaeplastida	Embryophyta	multi-	2n	g	7.3	-	10	39
2	*Spirodela polyrhiza*	Archaeplastida	Embryophyta	multi-	2n	g	0.082	-	47	46
1	*Dictyostelium discoideum*	Amoebozoa	Mycetozoa	alternates	1n	g	0.029	0.029	3	1
1	*Neurospora crassa*	Opisthokonta	Ascomycota	multi-	1n	g	3400	-	268	10,493
1	*Neurospora crassa*	Opisthokonta	Ascomycota	multi-	1n	s	-	0.60	10	90
5	*Saccharomyces cerevisiae*	Opisthokonta	Ascomycota	uni-	1n	g	0.39	0.35	68	475
9	*Saccharomyces cerevisiae*	Opisthokonta	Ascomycota	uni-	2n	g	0.23	0.23	392	3194
3	*Schizosaccharomyces pombe*	Opisthokonta	Ascomycota	uni-	1n	g	0.37	0.37	180	1308
1	*Marasmius oreades*	Opisthokonta	Basidomycota	multi-	2n	s	73	0.0038	40	111
1	*Schizophyllum commune*	Opisthokonta	Basidomycota	multi-	2n	g	20	-	17	9
1	*Schizophyllum commune*	Opisthokonta	Basidomycota	multi-	2n	s	-	0.02	24	300
4	*Caenorhabditis elegans*	Opisthokonta	Nematoda	multi-	2n	g	3.1	0.57	57	3553
1	*Caenorhabditis species*	Opisthokonta	Nematoda	multi-	2n	g	1.3	0.12	25	448
1	*Pristionchus pacificus*	Opisthokonta	Nematoda	multi-	2n	g	2.0	-	22	802
1	*Apis mellifera*	Opisthokonta	Arthropoda	multi-	1n	g	4.5	-	46	35
1	*Bombus terrestris*	Opisthokonta	Arthropoda	multi-	1n	g	3.9	-	32	23
1	*Chironomus riparius*	Opisthokonta	Arthropoda	multi-	2n	g	4.2	-	10	51
2	*Daphnia pulex*	Opisthokonta	Arthropoda	multi-	2n	g	3.1	-	30	1210
6	*Drosophila melanogaster*	Opisthokonta	Arthropoda	multi-	2n	g	5.1	0.13	175	3539
1	*Heliconius melpomene*	Opisthokonta	Arthropoda	multi-	2n	g	2.9	0.073	30	9
1	*Aotus nancymaae*	Opisthokonta	Chordata	multi-	2n	g	8.1	-	8	283
1	*Canis lupus*	Opisthokonta	Chordata	multi-	2n	g	4.5	-	4	27
1	*Chlorocebus aethiops*	Opisthokonta	Chordata	multi-	2n	g	9.4	-	3	8
1	*Clupea harengus*	Opisthokonta	Chordata	multi-	2n	g	2.0	-	12	19
1	*Ficedula albicollis*	Opisthokonta	Chordata	multi-	2n	g	4.6	-	7	55
2	*Gallus gallus domesticus*	Opisthokonta	Chordata	multi-	2n	s	-	0.91	6	384
1	*Gorilla gorilla*	Opisthokonta	Chordata	multi-	2n	g	11	-	1	83
13	*Homo sapiens*	Opisthokonta	Chordata	multi-	2n	g	12	0.17	3062	156,475
8	*Homo sapiens*	Opisthokonta	Chordata	multi-	2n	s	-	8.6	388	86,157
1	*Macaca mulatta*	Opisthokonta	Chordata	multi-	2n	g	5.8	-	14	307
3	*Mus musculus*	Opisthokonta	Chordata	multi-	2n	g	5.1	0.11	50	1614
2	*Mus musculus*	Opisthokonta	Chordata	multi-	2n	s	-	4.2	30	3697
3	*Pan troglodytes*	Opisthokonta	Chordata	multi-	2n	g	13	-	7	283
1	*Papio anubis*	Opisthokonta	Chordata	multi-	2n	g	6.2	-	12	475
1	*Pongo abelii*	Opisthokonta	Chordata	multi-	2n	g	17	-	1	51

Rates are averaged (mean) over all experimental estimates (unweighted). Rates are rounded to two significant digits. Color code: “supergroup” and “lower clade” columns are colored to highlight related clades; green saturation increases linearly with experiment counts in column “ct.”; a gradient from blue to red was applied across “Gbp^−1^ div^−1^” columns of Table 1 and Table 2, with blue indicating the lowest rates and red the highest. Abbreviations: ct. = number of independent estimates; g = germline; s = somatic cells; gen. = generation; div. = cell division; - = not determined.

**Table 2 cells-10-01224-t002:** Nuclear genome mutation rates from whole-genome experiments (MMR-deficient).

						Germ	Mutation Rates				MMR	
ct.	Species	Supergroup	Lower Clade	Cellularity	Ploidy	V. Soma	Gbp^−1^ gen.^−1^	Gbp^−1^ div.^−1^	Lines	Mutations	Efficiency	
2	*Arabidopsis thaliana*	Archaeplastida	Embryophyta	multi-	2n	g	810	27	14	8902	120 ^a^	100 ^b^
3	*Saccharomyces cerevisiae*	Opisthokonta	Ascomycota	uni-	1n	g	31	31	6	1840	79	89
4	*Saccharomyces cerevisiae*	Opisthokonta	Ascomycota	uni-	2n	g	13	13	25	3684	57	57
1	*Schizosaccharomyces pombe*	Opisthokonta	Ascomycota	uni-	1n	g	19	19	5	2597	51	51
2	*Caenorhabditis elegans*	Opisthokonta	Nematoda	multi-	2n	g	-	72	9	9110	-	130
1	*Gallus gallus domesticus*	Opisthokonta	Chordata	multi-	2n	s	-	47	2	6531	-	52

Rates are averaged (mean) over all experimental estimates (unweighted). Rates and correction efficiencies are rounded to two significant digits. Color code: “supergroup” and “lower clade” columns are colored to highlight related clades; green saturation increases linearly with experiment counts in column “ct.”; a gradient from blue to red was applied across “Gbp^−1^ div^−1^” columns of Table 1 and Table 2, with blue indicating the lowest rates and red the highest. Notes: ^a^ = efficiencies calculated from mutation rates per generation; ^b^ = efficiencies calculated from mutation rates per cell division. Abbreviations: ct. = number of independent estimates; g = germline; s = somatic cells; gen. = generation; div. = cell division; - = not determined.

**Table 3 cells-10-01224-t003:** List of whole-genome mutation rate experiments.

First Author	Year	Reference	Species	MMR Genotype
Lynch	2008	[2]	*Saccharomyces cerevisiae*	*WT*
Keightley	2009	[4]	*Drosophila melanogaster*	*WT*
Denver	2009	[5]	*Caenorhabditis elegans*	*WT*
Ossowski	2010	[6]	*Arabidopsis thaliana*	*WT*
Roach	2010	[12]	*Homo sapiens*	*WT*
Zanders	2010	[7]	*Saccharomyces cerevisiae*	*mlh1-7^ts^*
Nishant	2010	[59]	*Saccharomyces cerevisiae*	*WT*
Conrad	2011	[60]	*Homo sapiens*	*WT*
Denver	2012	[61]	*Caenorhabditis* species	*WT*
Ma	2012	[62]	*Saccharomyces cerevisiae*	*mlh1-7^ts^*
Kong	2012	[63]	*Homo sapiens*	*WT*
Ness	2012	[64]	*Chlamydomonas reinhardtii*	*WT*
Saxer	2012	[65]	*Dictyostelium discoideum*	*WT*
Sung	2012	[27]	*Chlamydomonas reinhardtii*	*WT*
Sung	2012	[27]	*Paramecium tetraurelia*	*WT*
Michaelson	2012	[66]	*Homo sapiens*	*WT*
Schrider	2013	[67]	*Drosophila melanogaster*	*WT*
Lang	2013	[68]	*Saccharomyces cerevisiae*	*WT*
Lang	2013	[68]	*Saccharomyces cerevisiae*	*msh2Δ*
Li	2014	[69]	*Homo sapiens*	*WT*
Keightley	2014	[70]	*Drosophila melanogaster*	*WT*
Stirling	2014	[71]	*Saccharomyces cerevisiae*	*WT*
Weller	2014	[72]	*Pristionchus pacificus*	*WT*
Serero	2014	[73]	*Saccharomyces cerevisiae*	*WT*
Serero	2014	[73]	*Saccharomyces cerevisiae*	*msh2Δ*
Zhu	2014	[74]	*Saccharomyces cerevisiae*	*WT*
Venn	2014	[75]	*Pan troglodytes*	*WT*
Meier	2014	[76]	*Caenorhabditis elegans*	*WT*
Behjati	2014	[17]	*Mus musculus*	*WT*
Lujan	2014	[40]	*Saccharomyces cerevisiae*	*WT*
Lujan	2014	[40]	*Saccharomyces cerevisiae*	*msh2Δ*
Jiang	2014	[22]	*Arabidopsis thaliana*	*WT*
Keightley	2015	[77]	*Heliconius melpomene*	*WT*
Francioli	2015	[78]	*Homo sapiens*	*WT*
Uchimura	2015	[79]	*Mus musculus*	*WT*
Baranova	2015	[80]	*Schizophyllum commune*	*WT*
Yang	2015	[24]	*Apis mellifera*	*WT*
Yang	2015	[24]	*Arabidopsis thaliana*	*WT*
Yang	2015	[24]	*Oryza sativa*	*WT*
Ness	2015	[81]	*Chlamydomonas reinhardtii*	*WT*
Farlow	2015	[82]	*Schizosaccharomyces pombe*	*WT*
Keith	2016	[83]	*Daphnia pulex*	*WT*
Rahbari	2015	[84]	*Homo sapiens*	*WT*
Haye	2015	[85]	*Saccharomyces cerevisiae*	*msh6Δ*
Behringer	2016	[86]	*Schizosaccharomyces pombe*	*WT*
Sharp	2016	[87]	*Drosophila melanogaster*	*WT*
Huang	2016	[88]	*Drosophila melanogaster*	*WT*
Sun	2016	[54]	*Schizosaccharomyces pombe*	*WT*
Sun	2016	[54]	*Schizosaccharomyces pombe*	*msh6Δ*
Smeds	2016	[89]	*Ficedula albicollis*	*WT*
Long	2016	[25]	*Tetrahymena thermophila*	*WT*
Blokzijl	2016	[18]	*Homo sapiens*	*WT*
Zámborszky	2017	[14]	*Gallus gallus domesticus*	*WT*
Watson	2016	[90]	*Arabidopsis thaliana*	*MSH2−/−*
Flynn	2017	[91]	*Daphnia pulex*	*WT*
Xie	2017	[33]	*Prunus persica*	*WT*
Xie	2017	[33]	*Prunus* hybrid	*WT*
Besenbacher	2016	[92]	*Homo sapiens*	*WT*
Hamilton	2017	[93]	*Plasmodium falciparum*	*WT*
Liu	2017	[94]	*Bombus terrestris*	*WT*
Ju	2017	[20]	*Homo sapiens*	*WT*
Krascovec	2017	[28]	*Bathycoccus prasinos*	*WT*
Krascovec	2017	[28]	*Micromonas pusilla*	*WT*
Krascovec	2017	[28]	*Ostreococcus mediterraneus*	*WT*
Krascovec	2017	[28]	*Ostreococcus tauri*	*WT*
Milholland	2017	[39]	*Homo sapiens*	*WT*
Milholland	2017	[39]	*Mus musculus*	*WT*
Feng	2017	[95]	*Clupea harengus*	*WT*
Maretty	2017	[96]	*Homo sapiens*	*WT*
Dutta	2017	[97]	*Saccharomyces cerevisiae*	*WT*
Jónsson	2017	[98]	*Homo sapiens*	*WT*
Pfeifer	2017	[99]	*Chlorocebus aethiops*	*WT*
Assaf	2017	[100]	*Drosophila melanogaster*	*WT*
Tatsumoto	2017	[101]	*Pan troglodytes*	*WT*
Schmid-Siegert	2017	[34]	*Quercus robur*	*WT*
Belfield	2018	[52]	*Arabidopsis thaliana*	*Atmsh2-1*
Meier	2018	[102]	*Caenorhabditis elegans*	*WT*
Meier	2018	[102]	*Caenorhabditis elegans*	*mlh-1*
Meier	2018	[102]	*Caenorhabditis elegans*	*pms-2*
Sharp	2018	[23]	*Saccharomyces cerevisiae*	*WT*
Krasovec	2018	[103]	*Silene latifolia*	*WT*
Thomas	2018	[104]	*Aotus nancymaae*	*WT*
Oppold	2017	[105]	*Chironomus riparius*	*WT*
Brody	2018	[16]	*Homo sapiens*	*WT*
Weng	2019	[106]	*Arabidopsis thaliana*	*WT*
Besenbacher	2019	[107]	*Pan troglodytes*	*WT*
Besenbacher	2019	[107]	*Gorilla gorilla*	*WT*
Besenbacher	2019	[107]	*Pongo abelii*	*WT*
Xu	2019	[108]	*Spirodela polyrhiza*	*WT*
Konrad	2019	[109]	*Caenorhabditis elegans*	*WT*
Krasovec	2019	[110]	*Phaeodactylum tricornutum*	*WT*
Koch	2019	[111]	*Canis lupus*	*WT*
Williams	2019	[112]	*Saccharomyces cerevisiae*	*WT*
Hanlon	2019	[35]	*Picea sitchensis*	*WT*
Hiltunen	2019	[31]	*Marasmius oreades*	*WT*
Lindsay	2019	[113]	*Mus musculus*	*WT*
Tian	2019	[19]	*Homo sapiens*	*WT*
Németh	2020	[15]	*Gallus gallus domesticus*	*WT*
Németh	2020	[15]	*Gallus gallus domesticus*	*MSH2−/−*
Orr	2020	[36]	*Eucalyptus melliodora*	*WT*
Bezmenova	2020	[32]	*Schizophyllum commune*	*WT*
Wang	2020	[26]	*Macaca mulatta*	*WT*
Wang	2020	[26]	*Neurospora crassa*	*WT*
Wu	2020	[114]	*Papio anubis*	*WT*
Wu	2020	[114]	*Homo sapiens*	*WT*
Sandler	2020	[115]	*Spirodela polyrhiza*	*WT*
Sandler	2020	[116]	*Lemna minor*	*WT*
Hofmeister	2020	[37]	*Populus trichocarpa*	*WT*
Sui	2020	[117]	*Saccharomyces cerevisiae*	*WT*
Zhou	2021	in review	*Saccharomyces cerevisiae*	*msh6Δ*

Experiments are arranged by publication date. A study with multiple measurements in the same species with the same MMR genotype is listed only once. More details about each measurement are available in Appendix A. Abbreviations: MMR = DNA mismatch repair.

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
