# Peer review of "Stability across the Whole Nuclear Genome in the Presence and Absence of DNA Mismatch Repair"

_cells, 2021, doi:10.3390/cells10051224_

Round 1
Reviewer 1 Report
This is a great recollection of previously reported genome mutation rates across different organisms. This recollection distinguishes between measurements performed using mismatch repair (MMR) proficient and MMR-deficient organisms, as well as other factors that affect mutation rates (ploidy and somatic vs germline). Certainly, this review will facilitate researchers interested in the DNA replication fidelity field to have access to this data. Moreover, the authors highlight interesting questions that remain open in the field and might be the focus of future investigations. In general, I strongly recommend publishing this review in Cells.
I have few comments that should be addressed and are listed as follows.
Page 2, second paragraph, the indicated rates for S. cerevisiae, D. melanogaster and c. elegans, etc, do not match the rates listed in Table 1. Is there a reason for that?
Page 2, "MMR gene MLH1 (mlh1-7ts. 3.7 Gbp-1 division -1)". Replace "." (after mlh1-7ts) with a semicolon.
Page 2, "(MMR-/MMR+=correction efficiency; 37/0.33=112)", from where comes the value "37"? should not be 3.7? If so, the correction efficiency is only 11 fold...
Page 3, "300 asexual divisions per sexual generation(26)." Space missing after the word "generation".
Page 5, "ranges: MMR-proficient with 0.23-0.91 8.1 Gbp-1 division-1;" What means the "8.1"?
Page 6, Table 2. Could you please explain the difference between the last two columns (below MMR efficiency).
Page 6, "reporter locus experiments are and inefficient" should read "reporter locus experiments are an inefficient"
Page 6, "a yeast genome is GC-poor, but certain ET-rich features". What is meant with ET-rich? perhaps AT-rich?
Page 6, "Tumors with mutations in with defects in DNA Polymerase (Pol)", should read "Tumors with mutations causing defects in DNA Polymerase (Pol).
Page 7, "Are MMR proteins depleted or in some other way impaired later in replication?". There is evidence that some of the MMR proteins are differentially expressed across the cell cycle, perhaps the authors could refer to a recent publication (PMID: 33303966) that present data supporting this idea. Also, the authors could cite PMID: 23622243 that showed that H3K36me3 promotes the recruitment of MutSalpha, supporting the idea that chromatin modifications might affect the recognition of mutations.
Page 9, At the end of the Table 3. "Zhou 2021 in revew", should read "in review"
Page 10, at the end of Supplementary Materials figure legend "from cited studies.." remove the last "."
Reviewer 2 Report
This review presents what will be a very useful compilation of mutation rate measurements and a discussion of the variation across organisms. Comments below are very minor and editorial in nature.
The authors might want to consider a different title – the focus is not really on MMR.
Please hyphenate “wild type” when it is used as an adjective. This also applies to other two-word adjectives such as “whole-genome” mutation rates, etc.
Suggest using 109 bp instead of Gbp throughout.
p., line 10 – whole “genome” mutation rates
“imputed” is not a commonly used word.
Please do not provide an abbreviation for a term used only once, e.g. “identity by descent”. Could abbreviate “wild type” throughout.
Section 3, line 6 – uses “assumed”
3 lines above Table 1 – if it to has?
Please explain the color coding in Tables 1 and 2 in the legends. The breaks in the headings under “mutation rates” in Table 1 need to be removed.
Section 4, paragraph 3 – delete “have been measured”
Section 6, 1st line – “an”. Line 5 – where was sequence complexity previously discussed? Do you mean discussed “below”? line 8 – define ET.
- 6, 4th line from the bottom – suggest changing “identity” to “presence” for clarity
Summary, line 5 - variation in MMR “landscapes” rather than “mechanisms”?
Table S1 – some extraneous numbers were in columns V-Z
Reviewer 3 Report
This is a very interesting review gathering, comparing, and critically evaluating information of whole genome mutation rates in multiple species. This information will uncover new insights into the patterns of mutagenesis across eukaryotes and beyond.
It would be useful if the authors included a critical evaluation of the methods used over the years to measure mutation rates. I would expect that technical improvements over the years must have made some results more reliable than others. I think these are important issues to address.
Minor point:
The titles of the columns in the tables are difficult to read.
